# Automatic Milk Quantity Recording System for Small-Scale Dairy Farms Based on Internet of Things

**Sanya Kaunkid** [1] , **Apinan Aurasopon** [2,*] **and Anut Chantiratiku** [3]

1   Department of Electrical Engineering, Faculty of Science and Technology, Nakhon Pathom Rajabhat University, Nakhon Pathom 73000, Thailand
2   Faculty of Engineering, Mahasarakham University, Mahasakham 44150, Thailand
3   Division of Animal Science, Faculty of Technology, Mahasarakham University, Mahasakham 44150, Thailand
*   Correspondence: apinan.a@msu.ac.th

**Abstract:** The milk quantity of dairy cows is the most important piece of data in farm management. However, it is difficult to measure and record the milk quantity for small-scale dairy farms. Therefore, the automatic milk quantity recording system for small-scale dairy farms is studied. It consists of a weight scale mechanism and an embedded system installed on a wheelbarrow for measuring and recording milk quantity. For the process of the system, the milk quantity of each cow is measured based on the load cell in kilogram units. The data such as real-time clock, cow ID, and individual and total milk quantity are recorded on a microSD memory card and sent based on the Internet of Things (IoT) for recording in a Google sheet. Furthermore, the system can alert the farmers to remove the teat cups when the cow milk comes to the end by detecting the derivative of milk quantity with respect to time. The experimental results show that the proposed system can correctly measure and record milk quantity. This system can help the farmers in improving and managing dairy farms effectively.

**Keywords:** small-scale dairy farms; load cell; embedded system; milk quantity; Internet of Things; Google sheet

## 1. Introduction

In the past decade, dairy farm automation has grown increasingly significant. The most essential feature of a smart farm is the use of IoT-based systems to monitor and operate the many sensor-based devices utilized on the farm. Connectivity facilitated by the internet between sensor systems permits the integration of previously difficult-to-achieve data streams. A cloud-based infrastructure is used to store the data collected from a variety of measuring equipment, including cattle collars, milking stations, and feed wagons, among others [1,2]. The cloud framework provides a perfect example of an appropriate method for minimizing the obstacles that must be overcome in order to carry out complete integration. The growth of technologies that allow for wirelessly connected sensor networks has made it possible to install sensor platforms at a low cost. These sensor platforms have uses in a variety of different industries and communities [3,4]. In the field of agriculture, especially small-scale dairy farms, the IoT and sensor-based technologies have been used to build technologies that help farmers make decisions and make farms run more efficiently. This saves them time and gives them information about the production process, the herd, and each animal.

Dairy milk contains many nutrients necessary for humans. It is an excellent source of protein, fat, lactose, vitamins, and minerals that are vital for body growth, health maintenance, and disease prevention [5,6]. Milk production in Thailand has increased rapidly, stimulated by population growth and dietary changes. In 2022, Thailand's domestic dairy cows have the capacity to produce raw milk at approximately 3800 tons per day from about 324,000 cows of 23,436 farms nationwide [7]. Milk yield contributes to 70% of the total milk from over 20,000 farms produced in the country and comes from small-scale

dairy farms based on family farms whose herds may range from two or three cows to over a dozen head.

The milk quantity is the most important factor in farm management where the total milk quantity of a dairy farm is used as the data for evaluating the business profits and distribution to the supplier. Meanwhile, the individual milk quantity produced per cow per milking session is the main criterion for assessing the worth of dairy cows and for culling decisions and mating decisions and is also important for properly adjusting feed amounts and formulating rations, in addition to estimating profitability [8]. For the big farms, using pipeline milking machines for cow milking, individual milk quantity is easily measured and recorded by a digital flow milk meter installed at each stanchion barn. Meanwhile, for small-scale dairy farms using bucket milking machines, it can be measured by using a flow milk meter, graduated glass milk flasks, or weighing and recording the results by hand. However, this leads to difficulties and an increase in milking time. In Mahasarakham province, 103 small-scale dairy farms are supervised by Khok Kao Cooperatives, which purchase raw milk from these farms at the agreed-upon price. The cooperative will process and sell raw milk to a number of product manufacturers. The average of the individual milk quantity of these farms is low at about 11.45 kg per day, while that of big farms is 18 kg per day [9]. Therefore, the cooperative is responsible for taking care of its member farms and promoting efficient dairy farming for cows with a high milk yield and low care costs. This study measured individual cow milk and recorded data based on the IoT, making it simple for cooperatives to access information about each farm. This innovation permits cooperatives to evaluate the efficiency of cattle rising and to supervise, recommend, and optimize the process for maximum efficiency.

Therefore, this research studies the automatic milk quantity recording system for a bucket milking machine based on the IoT. The milk quantity of each cow is measured in kilogram units and recorded on a microSD memory card and sent for recording in a Google sheet. The paper is organized as follows: Section 2 reviews the related work based on the IoT for small-scale dairy farms. Section 3 gives the materials and methods; the conventional method for measuring and recording the individual milk quantity; details of the proposed system; the control circuits; the flowchart for program processing implemented on an Arduino board. Section 4 shows the experimental results: milk quantity measurement, recording the data on microSD memory and a Google sheet, and the overmilking test. Section 5 discusses the work. Finally, Section 6 concludes the work.

## 2. Related Work

In the livestock area, milk production is obvious because it creates jobs and income and uses technology to make production more efficient. Technology may be used to regulate cow nutrition and milk production, as it does in several other industries, including industrial manufacturing, process management, and logistics. The phrase IoT is used to represent a sort of technology that is forecasted to have billions of products internet-connected and accessible from anywhere in the near future. Therefore, several IoT-enabled applications are feasible, such as smart parking, smart animal farms, and smart waste management systems. The IoT is now expanding in agriculture and livestock, enabling intelligent settings to comprehend their surroundings and interact with people. This section presents some initiatives related to technology helping in the management of cattle that are applied in precision agriculture based on IoT applied to small-scale daily farms. Table 1 shows the related studies in this research.

**Table 1.** Related studies in this research.

| References | Dairy Farm Scale | Data Extraction | Control and Notification System | IoT | Data Transmission | Solution |
|---|---|---|---|---|---|---|
| Memon et al. [10] | - | Sensors/automation | Fire alarm | Yes | WIFI and UDP | Control feeding, water supply, and gas depletion in farms; real-time agricultural information and recordkeeping |
| Zakeri et al. [11] | Small- and medium-sized farms | Sensors to monitor the quality of stored milk in tanks | Predictive decisions to maximize milk quality | - | Wired | System for proactive management of raw milk quality during the storage in a farm |
| Rodrigo, et al. [12] | Small-scale dairy farms | The milk sensors, RFID | Forecasting the milk production | Yes | HTTP | Combining data prediction and Internet of Things to manage milk production |
| Fouad, et al. [13] | Small-scale dairy farms | GPS | Data collected through the mobile application | Yes | Cellular | Mobile application for monitoring the productive and reproductive performance of small dairy herds. |
| Arago, et al. [14] | Small-scale commercial farms | Pan-tilt-zoom (PTZ) cameras/automation | Detects the cows and any estrus events | Yes | Long Term Evolution (LTE) | Smart Dairy Farming |
| Deniz et al. [15] | Small-and medium-sized farms | Acoustic signals | Classify three types of Events: chew, bite, and chew-bite | - | Zigbee protocol | Real-time detection and classification of events such as chew, bite, and chew-bite |
| Hwang et al. [16] | All farms | Single or multiple images of individual livestock | Diagnose livestock disease status | Yes | TCP/IP | Employing a mobile application based on Android OS to create cattle health anomaly information and transmit it to an expert for timely reaction and counseling. |
| Pimpa et al. [17] | - | Dataset of dairy cattle | Status of dairy cattle separated into 3 classes | Yes | HTTP | Dairy cattle health prediction: normal, possibly abnormal, and risk group |
| Anggraeni et al. [18] | - | Dataset of medical records | Diagnosis for three different cattle diseases | - | - | A mobile intelligent system for diagnosing cattle diseases and recommending first-aid actions. |
| Yan et al. [19] | Small-sized farms | Data of milk production | Data analytics in milk supply decision-making | - | - | The design of the tool for big data analytics can be applied in a cost-effective manner. |
| Gaworski et al. [20] | Small-sized farms | Fuel consumption | Mobile milking to calculate fuel usage. | - | - | Fuel use of the tractor that drives the mobile milking parlor |

A model to control feeding, water supply, and gas depletion in farms was presented [10]. It employs microcontrollers and actuators; water level, ultrasonic, gas, temperature, and humidity sensors; IP cameras. The model also includes an embedded system for real-time agricultural information and recordkeeping. This module comprises sensors and two microcontrollers, an Arduino Mega and Uno. The system has automated and manual modes. In automated mode, it uses thresholds as parameters to run the system. The animals are fed using a hopper with a valve, motor, and ultrasonic sensor to measure the food level. As food runs out, the hopper's valve opens to feed the animals.

Smart Farm is a model established by reference [11] to manage raw milk quality. They suggest that consumers' rejection of milk and other dairy products due to poor quality has major negative effects on the upstream stakeholders in the dairy supply chain. Smart Farm aims to provide milk producers and logistics service providers with next-generation automated systems based on interactive artificial intelligence to proactively monitor raw milk quality.

Rodrigo et al. proposed the MooCare model, designed to assist producers in the management of dairy cattle, in order to obtain better productivity indexes [12]. Using IoT devices, MooCare performs the automation and individualization of the animals feeding. Data gathered using such devices enable the MooCare scientific contribution, which consists of providing the milk production forecasting of each cow, using the ARIMA prediction engine. Thus, farmers can be notified in advance, enabling them to react in terms of designing a better nutritional plan, which can be individualized for each cow.

Fouad et al. developed a mobile application, MIT App Inventor, for monitoring the productive and reproductive performance of small dairy herds [13]. The mobile application

could be developed for continuous data collection and following-up small dairy farms in rural areas.

Arago et al. proposed a noninvasive and noncontact estrus detection system integrated IoT technology to improve the detection efficiency of standing-heat behaviors of cows. Pan-tilt-zoom (PTZ) cameras and a Python-driven Web Application improve the detection of cows in heat [14]. The dimensions of the barn are measured, and the FOVs of the cameras atop the cowshed are pre-calculated. Surveillance cameras identify cows and estrus events.

Deniz et al. [15] presented the design and execution of an electronic system created exclusively for real-time monitoring of dairy cow eating habits. The system is built on an embedded circuit that processes the sound emitted by the animal to identify, categorize, and quantify ruminant eating behavior events.

Hwang et al. [16] showed a livestock illness counseling system that works with an Android-OS smartphone. With this system, people can prevent the possible misuse of antibiotics and other medications for animals and check the status of livestock diseases early on so that they can be treated quickly.

This article [17] used artificial intelligence by presenting how artificial neural networks can be used to classify cows by how healthy they are. This helps farmers take care of their cows properly as healthy cows make good milk. The purpose of this study [18] was to describe the development of a mobile intelligent system for cattle illness detection and first-aid action recommendation. The system's central intelligence was built utilizing fuzzy neural networks.

This paper [19] solved issues in dairy supply chains and assisted dairy farmers, particularly small-scale producers, in utilizing data analytics in milk supply decision-making. This paper's objective [20] was to present and assess chosen technical features of mobile milking parlors, which might help in resolving the milking problem on some dairy farms.

## 3. Material and Methods

### 3.1. Small-Scale Dairy Farms

Most of the dairy farms in Thailand come from rural regions based on family farms or small-scale dairy farms, where the average size of a dairy herd is about 13–14 cows [8]. Most of these farms use the bucket milking machine for yielding milk because of the low cost and easy maintenance compared with the big farms using the pipeline milking machine system controlled by a computerized system [21].

### 3.2. Bucket Milking Machine

Figure 1 shows the bucket milking machine system that consists of four basic components: (a) a vacuum system, (b) pulsators that alter the vacuum level around the teat, (c) milking units made of four teat cups with liners connected to a claw, and (d) milk buckets. Upon attaching the teat cup, milking machines extract milk from the dairy cow by applying a partial vacuum to the teat, creating a pressure difference that results in the opening of the streak canal and milk flowing out from the teat cistern through a milk tube to a receiving bucket or container [21]. For milking processing, all cows stand at the stanchion barn, where the cows each have their own spot to eat and to be milked every day. One milking unit is used to milk a cow. For the example, as shown in Figure 1, the machine system has two milking units; therefore, it can milk two cows at the same time where N is the number of cows for milking in each time.

### 3.3. Recording Individual Milk Quantity

The milk quantity of dairy farms is the most important data in farm management. For the big farms using pipeline milking machines, the digital flow milk meters are installed at the stanchion barns to measure the milk quantity of each cow, which is automatically recorded by a computerized system. While small-scale dairy farms use bucket milking machines, the individual milk quantity can be measured by several methods, as shown

in Figure 2. Farmers must carefully read the scales and record the results by hand at the start and end of milking in each cow, using a scale flow milk meter or graduated glass milk flasks. Using weighing, after milking for each cow, the milk bucket has to be measured by weighing and calculated for the real milk weight. These methods are very difficult and cause the milking time to increase. While the digital flow milk meter can be used, however, it is very expensive. The record system was developed using smartphone application programming to aid in the management of small-scale dairy farms [22,23]. However, these recording systems still need the data on milk quantity from measurements.

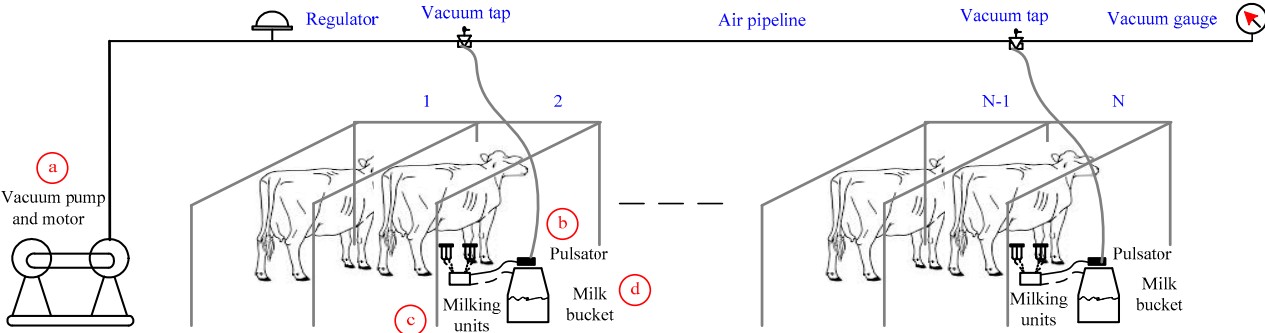

**Figure 1.** Bucket milking machine: two milking units.

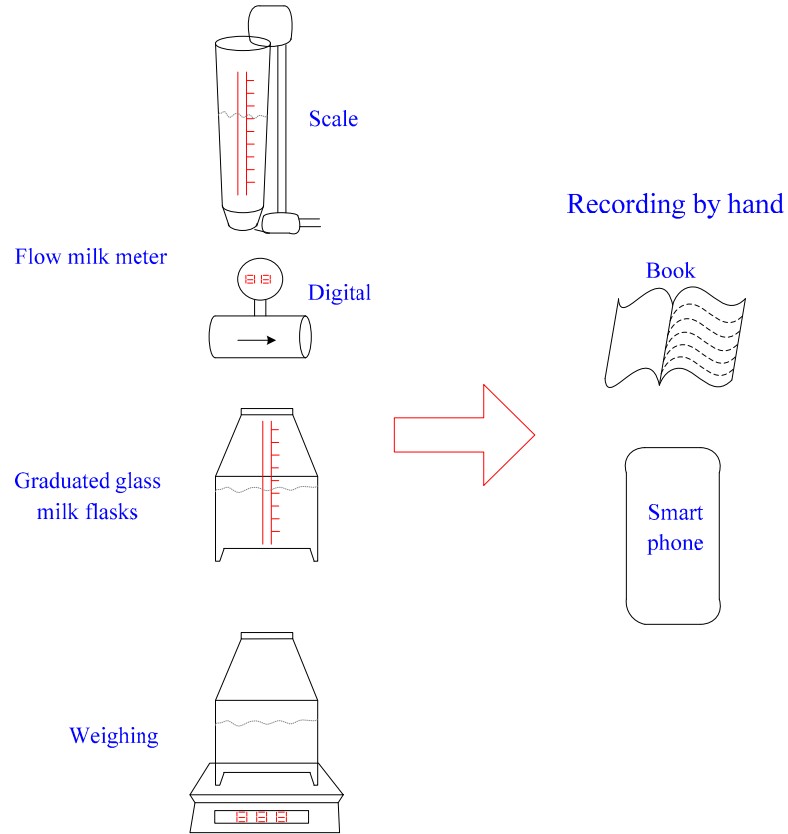

**Figure 2.** Milk quantity recording system by hand.

### 3.4. Proposed Systems for Automatic Milk Quantity Recording System

The objectives of this research are to measure and record the individual milk quantity. The proposed system should be simple and not affect the conventional milking process. In this section, we show how to design the system for achieving the research objectives. Figure 3 shows the basic block diagram of the proposed system. The main functions of the

control unit are to measure the milk quantity and record the data such as real-time clock, cow ID, and individual and total milk quantity in the recording system: microSD memory card and a Google sheet.

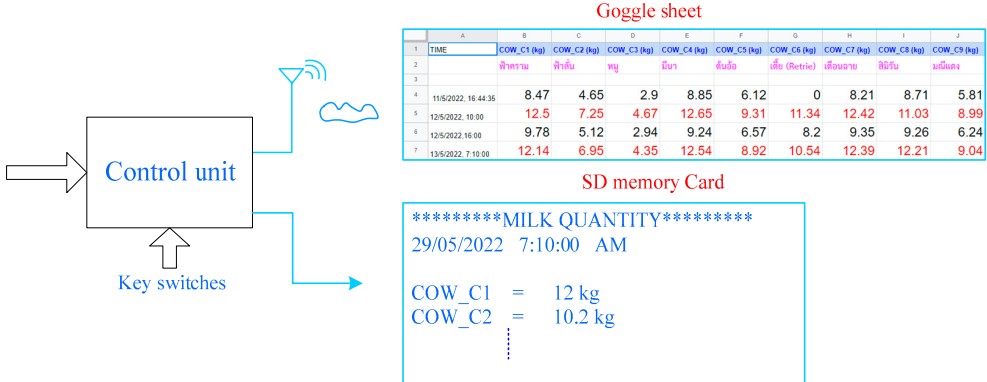

**Figure 3.** Proposed system.

### 3.5. Control Unit

The control unit for automatic milk quantity recording systems consists of the weight scale mechanism and the embedded system. The weight scale mechanism functions by loading the milk bucket while the embedded system processes the key switch input, milk quantity measurements, displaying the texts on an LCD and recording and sending the data to a Google sheet.

### 3.5.1. Weight Mechanism

From the bucket milking machine system as shown in Figure 1, the farmers generally use the stainless-steel milk bucket capacity available in 30 L; therefore, the total weight is about 38 kg including an empty milk bucket. When each cow is finished being milked, the farmer has to lift the milking unit to milk the other cows. This may be difficult for petite farmers. Therefore, the milk wheelbarrow is used, as shown in Figure 4, consisting of a steel structure and three wheels. It can hold up to 50 kg. Under the circle base, the load cell of 50 kg is installed and connected to the embedded system. When the milk bucket is put on the circle base, the resulting electrical signal is changed corresponding with the milk bucket weight and sent to the embedded system for processing the milk quantity.

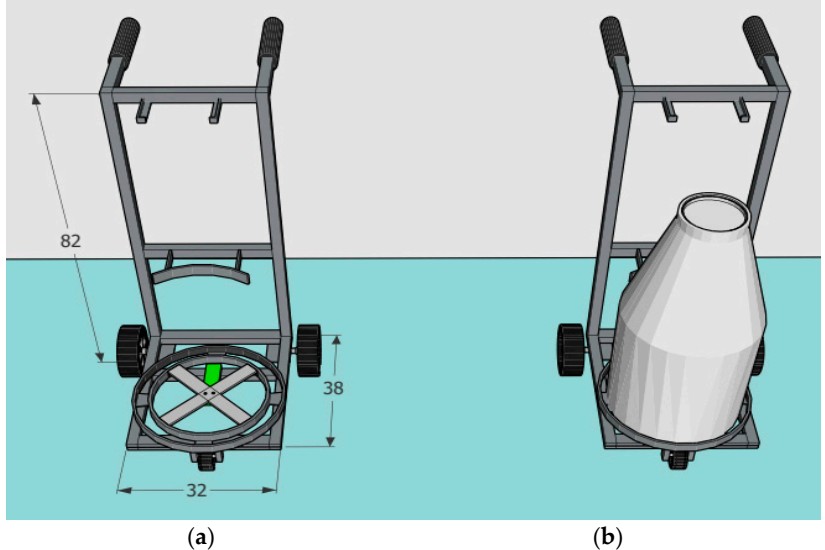

(**a**)　　　　　　　　　　　　　　　　　　(**b**)

**Figure 4.** Milk wheelbarrow: (**a**) weight scale mechanism and (**b**) milk bucket positioning.

### 3.5.2. Embedded System

We design the embedded system for the automatic milk quantity recording system by considering these factors: easy to use, small size, and robustness. In this subsection, details of the hardware and software are explained.

#### Hardware Design

The schematic circuit diagram of the automatic milk quantity recording system was developed, as shown in Figure 5. Such a system was designed based on a modular structure that consists of numerous electronics modules.

- The Arduino Mega 2560 PRO is utilized as the main controller. It is responsible for receiving data from the key switches to determine the cow number; collecting data from the load cell to measure milk quantity; storing the data that are the date, time, cow ID and individual and total milk quantity onto the microSD memory; sending all data to the Google Sheet platform through the Node MCU.
- The load cell converts force into an electrical signal that can be measured. The electrical signal varies accordingly with force applied. In order to measure weight, a scale is built using load cells that are wired to an HX711 amplifier and an Arduino. This experiment utilizes strain gauge load cells with a maximum weight capacity of 50 kg.
- The serial real-time clock (RTC) provides timekeeping features for use in a time-based system.
- The I2C LCD shows the weight quantity as well as the date and time taken from the RTC.
- A microSD card is used to store the information obtained from the reading of the sensor as well as the measurement time.
- Node MCU is used to establish a wireless connection and transmit all data to the Google Sheet platform.

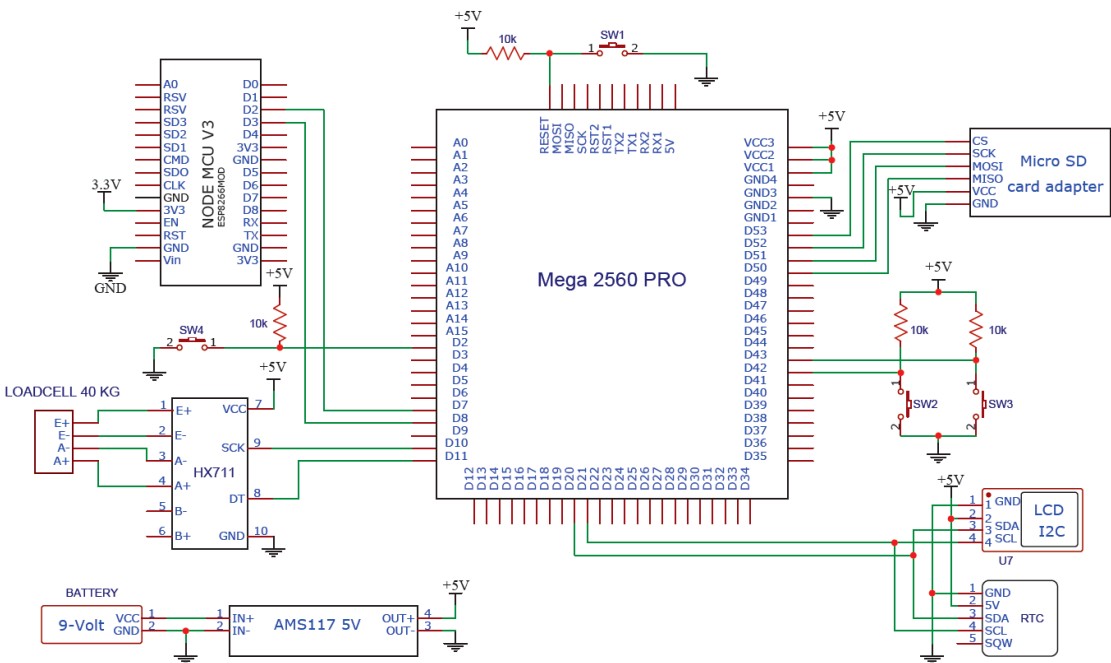

**Figure 5.** Circuit diagram for the proposed system.

#### Software Designing

Figure 6 shows the state transition diagram for the automatic milk quantity recording system. The system consists of several sub-transition states: initial state, receive state, process state, and transmit state. The initialization is started when powering on the system.

The initial state is shown on the main display. It transitions to the receive state by a switch "Cow number" to identify the number of cows such as C_1, C_2, . . . , and C_N. When it receives the cow number, this state transitions to the process state by pressing the "Enter" button. It starts measuring milk quantity. In the weight measurement processing, the zero setting technique uses the weight before the Enter switch is pressed to be set as zero kg.

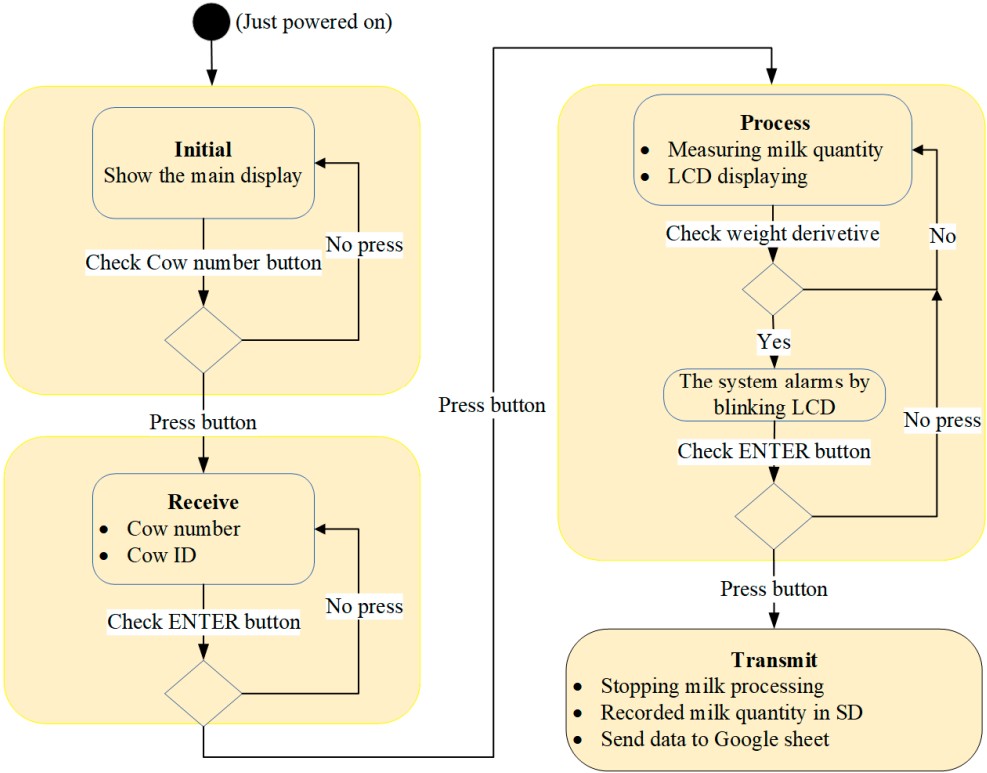

**Figure 6.** State transition diagram.

The result of the weight measurement processing and the milk quantity is displayed on the LCD in real time. While milk processing, the farmer has to observe the milk flow rate to lay out teacups to protect overmilking. Therefore, in order to help the farmer, the system checks the changing rate of milk weight with respect to time. If it goes to near-zero, the cow's milk comes to an end. The system alerts the farmer to press the "Enter" button to stop milk processing and lay out teacups. After the Enter button is pressed, this will transit to the transmit state, where the individual and total milk quantities are recorded in the SD memory card. After that, the system checks the cow number to see if all the cows are completely milked and sends the data for recording to a Google sheet. The algorithms in each state are implemented on the Arduino Mega 2560 PRO board by using an Arduino IDE.

Overmilking Detecting

Overmilking starts when the milk flow to the teat cistern is less than the flow out of the teat canal. It can cause an increase in the teat surface temperature, thickening of the cisternal wall, decrease in the teat diameter, and prolongation of the teat canal, and will, therefore, increase the possibility of bacteria entering the teat [24–26]. Therefore, the teat cup should be removed from the cow's udder before the overmilking starts. Many researchers have focused on the optimum switch-point for removing the milking unit or teat by detecting the flow rate [27,28]. However, this technique can only be applied in the automatic milking system. Therefore, to take care of the udder health of cows in small-scale farms, we propose a way to find the switch-point. Figure 7 shows a sketch of the milk

weight shown by the dashed line versus its derivative shown by the dotted line. The derivative of milk weight with respect to time can be found by

$$Mw' = \frac{dMw}{dt} \tag{1}$$

where $Mw$ is the milk weight. $dMw$ is the instantaneous rate of change of $Mw$. $dt$ is the instantaneous rate of change of time. $Mw'$ is the derivative of milk weight with respect to time.

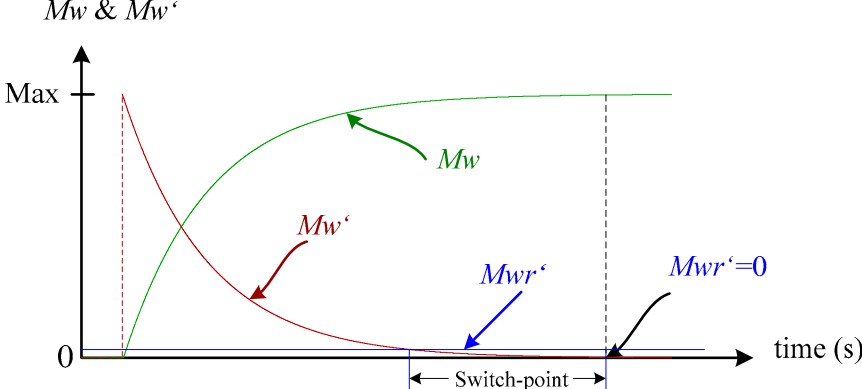

**Figure 7.** Milk weight and its derivative with respect to time.

To find the switch-point, Equation (1) is set as

$$Mw' \leq Mwr' \tag{2}$$

From Equation (2), the derivative reference, $Mwr'$, is used to set the switch-point. If the $Mwr'$ is set as zero, it means the switch-point found at the maximum of milk weight. To find the optimal switch-point, the $Mwr'$ can be found by experiments.

### 3.6. Recording System

When the milking processing is finished, the control unit sends the data: real-time clock, cow ID, and individual and total milk quantity, through the internet network for recording in a Google sheet. Some rural farms may lack internet access or have poor network quality. As a result, to avoid data loss, these files are also saved to SD memory cards. Figure 8 shows the recording format in the Google sheet and the SD memory card.

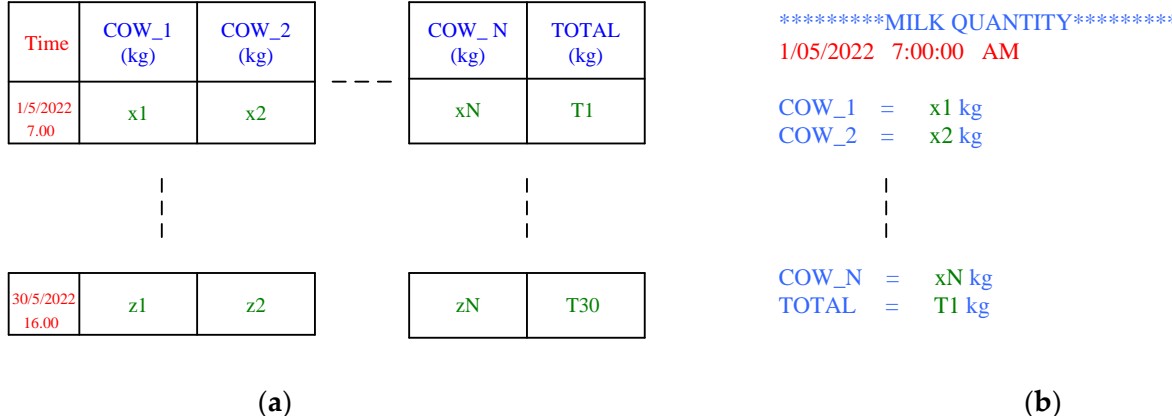

**(a)**            **(b)**

**Figure 8.** Data format: (**a**) Google sheet and (**b**) SD memory card.

## 4. Experimental Results

The proposed automatic milk quantity recording system for small-scale dairy farms based on the IoT was implemented and tested. As shown in Figure 9, the milk wheelbarrow was based on a steel structure and three wheels. The weight measuring mechanism was installed on the milk bucket base and the embedded system was put on the top. The milk wheelbarrow could hold up to 100 kg specified by three wheels; however, it was limited by the specification of the load cell capacity, 50 kg. From practical tests, the milk wheelbarrow was tested with a full bucket of milk, and the total weight was about 50 kg. The resulting wheelbarrow was robust and easy to use.

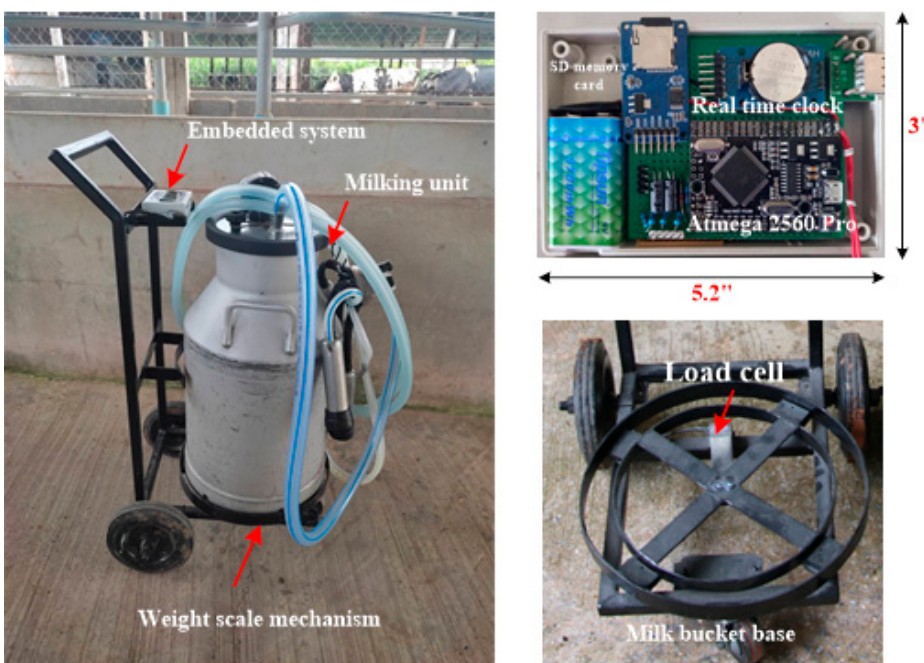

**Figure 9.** Prototype of the proposed system.

### 4.1. Weight Measuring Accuracy

For the first time of using the control unit for measuring the milk quantity, it has to be set-up. The zero and calibration factors can be found by the Auto calibration program [29]. In the first test, the weight test pendulums: 1, 5, 20, and 40 kg, were loaded. The results showed that its accuracy was more than 99.5% in all cases. For the second test, the control unit measured the milk quantity in the milking process. From the observation results, we found that the data for recording had an error. This is because of the vibration of the milk in the bucket. From the experiments, the number of times of reading the weight data from the ADC affected the correction of recorded data. Therefore, to protect this error, the number of times of reading the weight data from the ADC was tuned. From many experiments, the average of 10 data reading from the ADC gave the correct results.

### 4.2. Milk Quantity Measurement and Data Recording

The proposed system was tested at the small-scale dairy farm, which has 14 cows for yielding milk. The farm uses the bucket milking machine system and two buckets. It can only milk two cows at the same time. Figure 10 shows two rows of milking stalls, where each row has seven stanchion barns. Each cow has her name, which is defined corresponding to the cow ID such as C_1, C_2, . . . , and C_14. In the experiments, we used two control units for milking where each control unit can milk any cow.

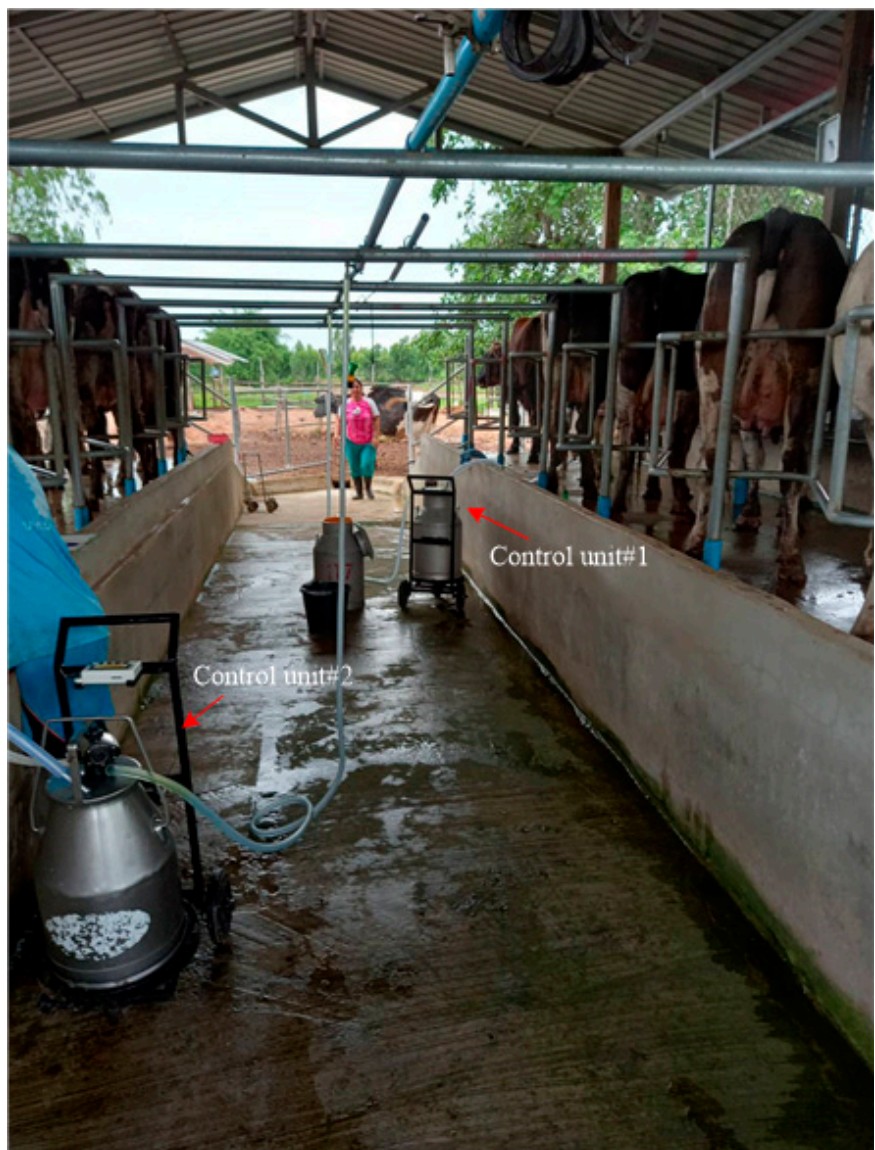

**Figure 10.** Milking stall used in experiments.

Figure 11 shows the experimental results on an LCD displaying two control units. The first, second, and third lines show cow ID, real-time weight, and total weight, respectively. After the user entered the cow ID and pressed the EN switch, it started the measuring mode. As the results show in Figure 11a, control unit#2 milked the first cow, C_3, while control unit#1 measured individual cow milk of C_4, C_6, and C_2 and was milking C_10, as shown in Figure 11b. When the cow milk of each cow came to the end, the LCD blinked to alert the user to press the EN switch to stop the measuring mode. When all cows finished milking, the data were recorded to the SD memory card and Google sheet, as shown in Figures 12 and 13, respectively.

Figure 12 shows the data recorded on the SD memory card of control unit#1 and control unit#2: end time of milking, cow ID, and individual and total weight. The data show that the milking position of each cow may be alternated in each time. These data were sent for recording in the Google sheet through NodeMCU 8266, which were sequenced as C_1, C_2, . . . , and C_14 by data processing in the Google sheet, as shown in Figure 13.

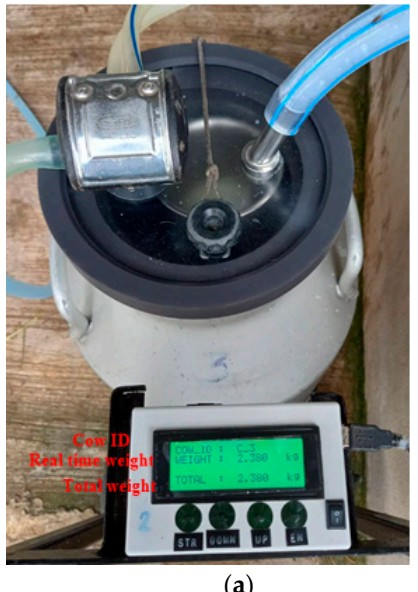 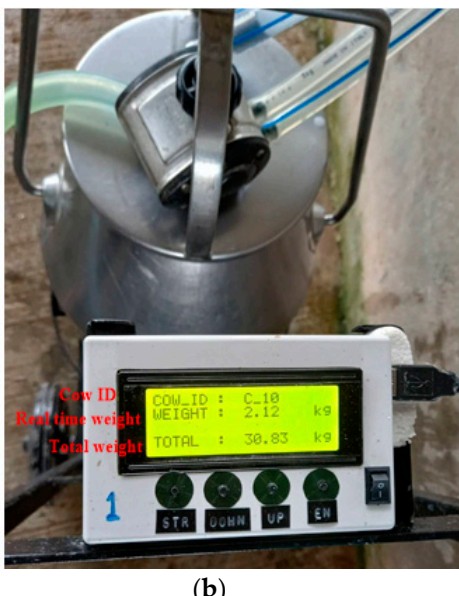

(**a**)             (**b**)

**Figure 11.** Real-time weight on LCD displaying (**a**) Control unit#2 and (**b**) Control unit#1.

| ********MILK QUANTITY******** | ********MILK QUANTITY******** |
|---|---|
| **********MACHINE#1********** | **********MACHINE#2********** |
| 12/05/2022    7:42:00    AM | 12/05/2022    7:45:00    AM |
| COW_C4    =    11.65    kg | COW_C3    =    4.53    kg |
| COW_C6    =    11.34    kg | COW_C1    =    12.5    kg |
| COW_C2    =    5.73    kg | COW_C5    =    9.31    kg |
| COW_C10    =    11.71    kg | COW_C7    =    12.42    kg |
| COW_C11    =    5.76    kg | COW_C8    =    11.03    kg |
| COW_C12    =    10.8    kg | COW_C9    =    8.99    kg |
| COW_C13    =    12.55    kg | COW_C14    =    7.48    kg |
| Total    =    69.54    kg | Total    =    66.26    kg |

(**a**)             (**b**)

**Figure 12.** Data recording on SD memory card: (**a**) Control unit#1 and (**b**) Control unit#2.

| TIME | COW_C1 (kg) ฟ้าคราม | COW_C2 (kg) ฟ้าลั่น | COW_C3 (kg) หมู | COW_C4 (kg) มีนา | COW_C5 (kg) ต้นอ้อ | COW_C6 (kg) เตี้ย | COW_C7 (kg) เดือนฉาย | COW_C8 (kg) ลินจีรัน | COW_C9 (kg) มณีแดง | COW_C10 (kg) ขวัญแก้ว | COW_C11 (kg) วันวิสาห์ | COW_C12 (kg) ร่องคำ | COW_C13 (kg) ส้มโอ | COW_C14 (kg) ยิปซี | Total (kg) | |
|---|---|---|---|---|---|---|---|---|---|---|---|---|---|---|---|---|
| 11/5/2022, 16:44:35 | 8.47 | 4.65 | 2.9 | 8.85 | 6.12 | 0 | 8.21 | 8.71 | 5.81 | 7.81 | 3.21 | 7.93 | 8.14 | 4.82 | 85.64 | บ่าย |
| 12/5/2022, 7:02:00 | 12.5 | 5.73 | 4.53 | 11.65 | 9.31 | 11.34 | 12.42 | 11.03 | 8.99 | 11.71 | 5.76 | 10.8 | 12.55 | 7.48 | 135.8 | เช้า |
| 12/5/2022, 16:00:32 | 9.78 | 5.12 | 2.94 | 9.24 | 6.57 | 8.2 | 9.35 | 9.26 | 6.24 | 8.77 | 2.88 | 7.77 | 8.34 | 5.2 | 99.66 | บ่าย |
| 13/5/2022, 7:10:00 | 12.14 | 6.95 | 4.35 | 12.54 | 8.92 | 10.54 | 12.39 | 12.21 | 9.04 | 10.47 | 6.64 | 9.92 | 12.33 | 7.51 | 135.95 | เช้า |
| 13/5/2022, 16:45:02 | 8.67 | 4.43 | 2.89 | 9.81 | 6.71 | 6.49 | 8.54 | 9.41 | 6.61 | 6.51 | 3.79 | 7.55 | 8.24 | 5.65 | 95.3 | บ่าย |
| 14/5/2022, 7:10:18 | 12.12 | 7.76 | 4.3 | 13.04 | 4.54 | 5.22 | 12.32 | 12.04 | 8.7 | 10.38 | 4.85 | 10.73 | 12.29 | 7.62 | 125.91 | เช้า |
| 14/5/2022, 16:56:50 | 8.9 | 4.79 | 3.43 | 9.83 | 5.17 | 6.55 | 8.52 | 9.45 | 6.41 | 7.84 | 3.41 | 7.26 | 8.54 | 5.43 | 95.52 | บ่าย |
| 15/5/2022, 7:10:00 | 10.97 | 7.01 | 9.38 | 13.03 | 8.68 | 9.32 | 11.98 | 13.16 | 8.7 | 11.37 | 7.35 | 10.62 | 0.08 | 6.58 | 128.22 | เช้า |

**Figure 13.** Data recording in Google sheet.

*4.3. Overmilking Test*

Overmilking can cause udder health problems in cows. Therefore, the milking unit should be removed from the udder when cow milk nearly comes to the end [26,27]. To find the optimum switch-point for removing the milking unit, the derivative of cow milk quantity with respect to time was detected. Two parameters, the time interval and derivative reference, $Mwr'$, were observed. The time interval should be as low as possible. However, it is limited by the time of program processing and the ability of the microcontroller. From the experiments, the time interval was set to the lowest of 1 s. Figure 14 shows the results of the serial plotter of the Arduino IDE on a plot of milk weight change of cow, C_10, versus time. The maximum of milk weight was 11.71 kg and its steady-state time was 260 s. From the test, the derivative reference, $Mwr'$, was observed as 0 g. The results show that the switch-point was found by 262 s. The system alerted by LCD blinking. It was delayed by 2 s, because the program loop was set as two. However, at this time, it did not affect the health of the cow udder. The switch-point can be changed by varying the derivative reference, $Mwr'$.

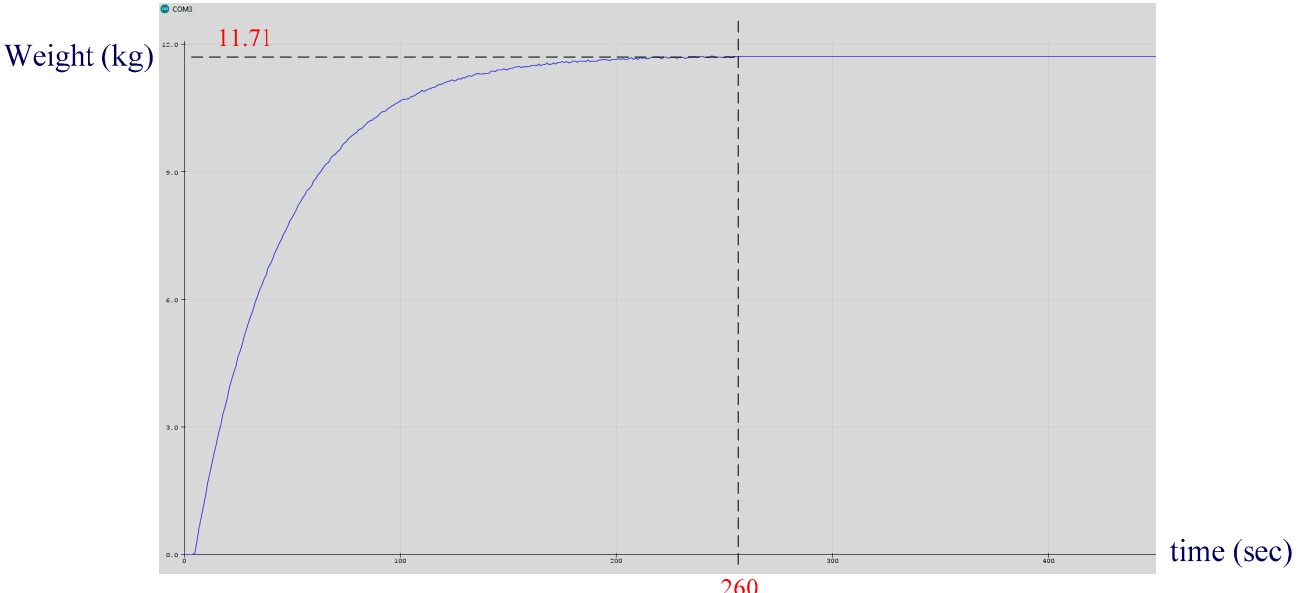

**Figure 14.** Milk weight change of cow, C_10, versus time plot.

## 5. Discussions

This research studied the automatic milk quantity recording system for small-scale dairy farms. The designed system was built from the prototypes, which consisted of the weight scale mechanism and the embedded system installed on the wheelbarrow for measuring the milk quantity and recording the data in a Google sheet. For measuring the milk quantity in the milking process, the proposed system was accurate and close to that of the conventional weighting method. Moreover, the process of the proposed system is simple, easy to use, and does not affect the conventional milking process [21].

To prevent overmilking from causing udder health problems in cows, a digital flow milk meter to detect the milk flow rate is used; however, this technique has only been used on large farms controlled by computerized systems [27,28]. For small-scale dairy farms with low cost, a new technique, detecting the derivative of milk weight with respect to time, was proposed. The experimental results showed that the proposed system can detect the switch-point and alert the farmers to remove the teat cups before the overmilking starts.

Because of the construction of the proposed system, the weight scale mechanism and the embedded system were installed on the wheelbarrow. It can easily be moved to milk the cows at each stanchion barn. Moreover, the proposed system can be applied to mobile

milking parlors [20] or portable milking machines [30] for small-scale dairy, goat, or sheep farms on pasture conditions.

## 6. Conclusions

This paper designed and implemented an automatic milk quantity recording system for small-scale dairy farms. The proposed system consists of the milk wheelbarrow where the weight scale mechanism is installed on the circle base for placing the milk bucket. This can make the farmers easily transfer the milk bucket to each stanchion barn, while the embedded system for measuring the milk quantity is placed on the top of the milk wheelbarrow. It is small in size, compact, and easy to use. To verify the proposed system on the small-scale dairy farm, two milking units and 14 cows were tested by two prototypes.

From the experimental results, the proposed system was tested by measuring the weight test pendulums of 1, 5, 20, and 40 kg. The results showed that it had an accuracy of more than 99.5 percent. When it was used in the milking process, the results showed that it can correctly measure the individual and total milk quantity by comparing with the digital weighing. Its precision exceeded 99 percent. During the overmilking tests, the system could alert the farmers to remove the teat cups when the cow milk came to an end. It was delayed by about 2 s because of the time limit of the program. For the recording system, the data, such as real-time clock, cow ID, and individual and total milk quantity, were correctly recorded on the microSD memory card and Google sheet. These processes help the farmers improve and manage dairy farms effectively. Furthermore, the Khok Kao Cooperatives can access the information about each dairy farm, resulting in evaluating, supervising, recommending, and optimizing the process of each dairy farm for maximum efficiency.

**Author Contributions:** Conceptualization, A.A. and A.C.; methodology, A.A.; software, A.A.; validation, S.K., A.A. and A.C.; formal analysis, A.A.; investigation, S.K. and A.A.; resources, A.A.; data curation, A.A.; writing—original draft preparation, A.A.; writing—review and editing, S.K. and A.A.; visualization, A.A.; supervision, A.A.; project administration, A.A. and A.C.; funding acquisition, A.A. All authors have read and agreed to the published version of the manuscript.

**Funding:** This research is financially supported by Mahasarakham University.

**Institutional Review Board Statement:** Not applicable.

**Data Availability Statement:** The data presented in this study are available on request from the Corresponding author.

**Acknowledgments:** This research is financially supported by Mahasarakham University, and we would like to thank the Khok Kao Cooperatives for their support with our experiments.

**Conflicts of Interest:** The authors declare that they have no conflict of interest to report regarding the present study.

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
