# Peer review of "Automatic Milk Quantity Recording System for Small-Scale Dairy Farms Based on Internet of Things"

_agriculture, doi:10.3390/agriculture12111877_

Round 1

Reviewer 1 Report

Although the Authors wrote what is the content of the article (proposal regarding the recording system ...), in my opinion it would be worth saying that "The purpose of the work / research / study was ...". Before formulating the research goal, it would be worth presenting the research problem. I think that on the basis of the review of the state of knowledge in Introduction, one can present a research problem, as well as a gap in the current state of knowledge. Besides, I suggest that you write down what was the cognitive (scientific) goal and what was the utilitarian (useful) goal of the presented research study. The content of the article refers mainly to the achievement of the utilitarian goal, however, the authors also achieved the scientific goal and it is worth mentioning.

In my opinion, it would be useful to provide more information on the structure of dairy production in Thailand in the Introduction. Regardless of the data already provided, it is worth writing down how many dairy farms there are, what is the average size of a dairy herd, what is the average milk yield of cows and what are the trends in this regard. Thanks to this, it will be easier for the authors to justify the importance of the topic discussed and developed in the article. If smaller dairy farms (with a small number of cows) dominate in Thailand, then the more valuable and sensible it is to implement the solution presented by the authors of the article.

The authors gave the acronym IOT as a full extension of the concept of "Internet of Things" (lines: 42, 45). However, in the next paragraph, the acronym IoT was used (lines: 54, 56, 58, 61). Is IoT the same as IOT? I guess so, so it would be worth standardizing the spelling of the acronym.

In Table 1, in the first column, you need to add numbers to the cited articles, which will arrange the articles in the correct order in the References.

In Figure 1, I am able to guess why the designations 1 and 2 were used. However, I cannot interpret the designations N-1 and N. Such details should be completed in the caption to the Figure.

I think that the principle of manual measurement of the amount of milk dispensed in Figure 2 could be described in more detail in section 3.2. For this purpose, it would be worth using the detailed words presented in Figure 2. Thanks to this, it would be easier for the reader to imagine the principle of operation of the presented measurement system.

On line 224, you must write "hardware" instead of "hard ware".

For the sake of order, in the description of the formula (1), the interpretation "dt" must be given.

Some sentences require linguistic correction and the selection of the right words to describe the situations under consideration. For example, in line 366 the authors write that the milking stall has two rows. Stall in this case means a milking stall, so you can write about two rows of milking stalls, as shown in the photo (Figure 10).

At the end of the article, there is no discussion of research results, which should be included in scientific publications. In this discussion, it is worth pointing to the essence of developing research taking into account the measurement of the amount of milk and to compare your own device with other solutions for measuring the amount of milk, taking into account the accuracy of the measurement, the level of automation of the work task and other aspects presented in the specialist literature on the subject. For example, it would be useful to expand on the question of where to use the milk meter. This could be a stationary milking installation in the barn. It is worth mentioning that the device could be used to measure the amount of milk in a milking installation in pasture conditions, where the cows are kept on pasture. Such a mobile milking machine installation, intended for dairy farms with a small herd of dairy cows, is presented in the article "Improvement of mobile milking parlours in small dairy farms including technical and functional aspects". By quoting this material, it would be possible to additionally justify the importance of the idea of ​​measuring the amount of milked milk presented in the article, not only in a cowshed, but also under conditions of milking cows on pasture, on farms with a small scale of production. Thus, it would be possible to emphasize the importance and value of the presented technical solution and the possibility of extending its use, not only in the cowshed.

The way of presenting publications in References must be adapted to the editor's requirements.

Author Response

We are grateful to the reviewers for their insightful comments on my paper. We have been able to incorporate changes to reflect most of the suggestions provided by the reviewers. We have highlighted the changes within the manuscript. Here is a point-by-point response to the reviewers' comments and concerns.

Reviewer 2 Report

1. The problem statement and motivation of the study missing in the abstract. How the proposed automatic milk quantity recording system is better than the previous one? It needs to be discussed with numerical findings.

2.  In the introduction section, from lines 35-39, the authors have stated the mechanism followed by small and big farms for measuring the milk quantity. However, the authors fail to identify the problem in both farms, which could provide an opportunity to conduct the study scientifically. The problem with proper statistics needs to present in the introduction section.

3. The motivation, limitations, and contributions must be incorporated in the introduction section.

4. The presentation of the proposed architecture can be improved, to conclude that it is based on the Internet of Things.

5. Before the conclusion section, a new section needs to be created to discuss all the results in brief for better readability.

6. The numerical findings observed in the study can be included in the conclusion and abstract of the study.

7. A comparative table needs to be present to justify whether the proposed system is better than the previous one.

Author Response

(The authors gave the same response as above.)

Reviewer 3 Report

The study may be interesting to investigate an automatic milk quantity recording system for small-scale dairy farms. The authors proposed system consists of a weight scale mechanism and an embedded system installed on the wheelbarrow for measuring and recording milk quantity. The paper discusses technical issues with the automatic milk quantity recording system. But scientific values are not clearly presented in this paper. Thus, the authors should explain the research questions and what they achieved for science.

In addition, I have some suggestions:

1.     The Introduction refers to some studies but does not explore the specifics of the use of the Internet of Things. It would be valuable to explain the advantages of the Internet of Things for small-scale dairy farms.

2.     In the Related Works section, the authors discuss only 5 works. Authors should provide more in-depth research of related papers.

3.     The methodology should be described in detail and transparent. Also, the authors should be given access to the data for the possibility of repeated experiments. It would be better to include some statistical analysis for an explanation of observed documents.

4.     The authors should consider the possible limitations they face in the research. It would also be useful to state the reason why the authors decided to use the methodology described.

5.     At the end of the paper, the authors give answers to the questions they asked in the introduction. However, the answers sound somewhat unscientific. The authors should add more arguments and facts to support their answers. This would give more scientific value to the paper.

6.     The conclusion section should be extended using: the results of the research, a discussion of related research, and a comparison between the authors’ results and the initial hypothesis.

Author Response

(The authors gave the same response as above.)

Round 2

Reviewer 1 Report

Thank you for taking into account the suggestions and comments presented in the review of the article. 

Reviewer 3 Report

The paper can be accepted in present form